# *Torenia sp.* Extracts Contain Multiple Potent Antitumor Compounds with Nematocidal Activity, Triggering an Activated DNA Damage Checkpoint and Defective Meiotic Progression

**DOI:** 10.3390/ph17050611

**Published:** 2024-05-10

**Authors:** Qinghao Meng, Robert P. Borris, Hyun-Min Kim

**Affiliations:** 1School of Pharmaceutical Science and Technology, Tianjin University, Tianjin 300072, China; 2Division of Natural and Applied Sciences, Duke Kunshan University, Kunshan 215316, China

**Keywords:** *Torenia species*, DNA repair, meiosis, germline development, medicinal plants, herbs

## Abstract

Previously, we analyzed 316 herbal extracts to evaluate their potential nematocidal properties in *Caenorhabditis elegans.* In this study, our attention was directed towards *Torenia sp*., resulting in reduced survival and heightened larval arrest/lethality, alongside a noticeable decrease in DAPI-stained bivalent structures and disrupted meiotic progression, thus disrupting developmental processes. Notably, *Torenia sp*. extracts activated a DNA damage checkpoint response via the ATM/ATR and CHK-1 pathways, hindering germline development. LC–MS analysis revealed 13 compounds in the *Torenia sp*. extracts, including flavonoids, terpenoids, tanshinones, an analog of resveratrol, iridoids, carotenoids, fatty acids, and alkaloids. Of these, 10 are known for their antitumor activity, suggesting the potential of *Torenia species* beyond traditional gardening, extending into pharmaceutical and therapeutic applications.

## 1. Introduction

The market for medicinal foods is experiencing rapid growth, anticipated to reach a yearly value of USD 2.1 billion by 2030, with an estimated expansion rate of 5.2% from 2022 [1]. While these foods are effective, it is vital to acknowledge that numerous herbs can have toxic properties, potentially resulting in adverse effects. Therefore, comprehending the mechanisms of herbs and recognizing the potential risks linked to their usage is imperative.

To evaluate the potential toxicity of herbal drugs, researchers have utilized various model systems. *Caenorhabditis elegans* (*C. elegans*) presents several advantages owing to its simple yet effective system facilitating the understanding of complex processes [2,3,4]. With a substantial genetic similarity to humans, conserved major biological pathways, and a wide array of genetic tools including transgenic models, gene knockouts, and RNAi depletions, *C. elegans* offers an invaluable platform for such studies [5,6].

In our previous study, we examined 316 herbal extracts to assess their potential antitumor effects in *C. elegans* [7]. Among the extracts, 16% showed decreased survival rates and larval arrest or lethality, suggesting that larval arrest contributes to worm viability outcomes. Interestingly, a minority of herb extracts revealed a noteworthy occurrence of males, a decrease in DAPI-stained bodies, and irregularities in meiotic progression, indicating abnormal meiotic development following herbal treatment.

In this report, we focused on *Torenia sp*. When *Torenia species (Torenia sp*.) extracts were administered to worms, it triggered a DNA damage checkpoint response through the ATM/ATR and CHK-1 pathways. This resulted in hindered germline development and pachytene apoptosis mediated by DNA damage. These findings suggest that *Torenia sp*. extracts hinder the process of DNA damage repair. Furthermore, our investigation unveiled a noticeable decrease in DAPI-stained bivalent structures and disrupted meiotic progression, further underscoring the disruptive effects of *Torenia sp*. extracts on developmental processes. Additionally, our mass spectrometry analysis revealed the presence of 13 compounds in the *Torenia sp*. extracts, with 10 of them known for their potential antitumor activity. Interestingly, 12 of these compounds overlap with those found in *Onobrychis cornuta*, a plant known for its strong potential as an antitumor herb. This highlights the potential broader therapeutic implications of *Torenia species* beyond gardening.

## 2. Results

### 2.1. Torenia sp. Extracts Exhibit Nematocidal Potency

The herbal extracts derived from *Torenia species* (*T.s.*) displayed markedly diminished survivability in comparison to the untreated control group, and a reduction was comparable to that of *Onobrychis cornuta* (*O.c.*) that we reported previously (Figure 1A, 89 vs. 41 in +DMSO and *O.c.*-B; 89 vs. 38 in +DMSO and *T.s.*-A at 0.03 µg/mL of herb extracts). Also, worms exposed to *Torenia sp*. extracts demonstrated larval arrest and/or lethality (Figure 1A, 91 vs. 43 in +DMSO and *O.c.*-B; 91 vs. 45 in +DMSO and *T.s.*-A at 0.03 µg/mL of herb extracts). This observation suggests that a link between compromised survivability is likely due to growth defects.

In *C. elegans*, mistakes occurring during the segregation of sex chromosomes may cause offspring to possess abnormal sex chromosome compositions, leading to a higher occurrence of males, referred to as the HIM (high incidence of males) phenotype. This phenotype serves as a marker for investigating the effects of such factors on sex chromosome segregation and chromosomal abnormalities [3,8]. *Torenia sp*. (*T.s.*) extracts also induced a significant increase in HIM, presenting potential sex chromosome mis-segregation and abnormal meiotic development (0.3 vs. 4 in +DMSO and *O.c.*-B; 0.3 vs. 4 in +DMSO and *T.s.*-A at 0.03 µg/mL of herb extracts).

### 2.2. Dose-Dependent Nematocidal Effects

To further confirm the nematocidal effects, we examined whether different doses of *Torenia sp*. were associated with the observed phenotypes. With a ten-fold increase in the dosage of the herbal extract, we noted a proportional decrease in survivability for both herbal extracts, demonstrating a distinct dose-dependent phenomenon (Figure 1A). For instance, in *Onobrychis cornuta*-B, survivability changed from 41, 23, to 19, and in *Torenia sp*. (*T.s.*)-A, it changed from 38, 29, to 24 for concentrations of 0.03, 0.3, and 3 µg/mL, respectively.

The number of adults also decreased with increasing herb concentrations, indicating that the herb extracts indeed inhibited mitotic growth. For instance, in *T.s*.-A, the percentage of adults ranged from 45% to 40% and then to 31% for concentrations of 0.03, 0.3, and 3 µg/mL, respectively.

While we observed the induction of the HIM phenotype with *O.c.*-B and *T.s*.-A (Male %), our observations did not reveal a distinct dose-dependent relationship, indicating that the observed defective sex chromosomal segregation may not be directly linked to the varying dosages of the herb extracts administered (Figure 1A). In *O.c*-B, the HIM phenotype was observed at 4, 4, and 3.8, and in *T.s.*-A, it was observed at 4, 5.5, and 3.8 for concentrations of 0.03, 0.3, and 3 µg/mL, respectively. Also, it is important to note that *T.s*. extracts dissolved in either water or hexane induced the HIM phenotype compared to the wild type (4% in *T.s.*-A; 3.2% in *T.s.*-H at 0.03 µg/mL). Subsequently, due to the shortage of *T.s.*-A, further assays were conducted exclusively using *T.s.*-H.

### 2.3. The Influence of Torenia species Extracts on Nematode Survival and Bacterial Growth

The reduced survival of *C. elegans* could be attributed to diminished bacterial growth rather than a direct association between *C. elegans* and *T.s.* extracts. To examine this hypothesis, we explored whether *T.s.* extracts hinder bacterial growth. No notable deficiency in bacterial growth was detected during either 10 or 24 h of incubation, suggesting that bacterial growth did not significantly contribute to the nematocidal effects (Figure 1B, 0.13 vs. 0.14 in op50 + DMSO and op50 + *T.s.*-H, *p* = 0.1 at 24 h of incubation; 0.10 vs. 0.10 in op50 + DMSO and op50 + *T.s*.-H, *p* = 0.1 at 10 h of incubation).

Collectively, our findings suggest that extracts from *Torenia species* impede survivability, larval growth, and proper sex chromosomal segregation. Additionally, the dosage of herb extracts appears to correlate with survivability and larval arrest.

### 2.4. Herbal Extracts Induce Abnormal Germline Development

In *C. elegans*, nuclei are spatially and temporally arranged along the progression of germline development. Mitotic nuclei are found at the distal end of the premeiotic tip (PMT), transitioning to meiotic prophase as they move away, where nuclei adopt a crescent shape in the transition zone [3,9].

Since *Torenia species* extracts caused defective meiotic progression, we explored their impact on germline development. Adult hermaphrodites were dissected, stained with DAPI, and examined. In the control group, nuclei exhibited orderly arrangement during germ cell development, whereas worms exposed to *Torenia species* extracts displayed wider gaps between nuclei at the PMT-TZ and pachytene stages, suggesting an abnormal advancement of germline nuclei (Figure 2A). Quantitative analysis of the distance between adjacent nuclei validated that *Torenia species* extracts caused notable spatial disarray of nuclei at both the PMT and pachytene stages (Figure 2B). In the PMT stage, the distances were 4.2 vs. 7.4 µm in the control and *Torenia species*, *p* < 0.0003. In the pachytene stage, the distances were 5.3 vs. 7.6 µm in the control and +*T.s*., *p* < 0.0003, as determined by the two-tailed Mann–Whitney test.

Crescent-shaped nuclei are indicative of the transition from the mitotic to meiotic stages [3,10]. The herb extracts did not alter the occurrence of crescent-shaped nuclei in PMT (Figure 2B, *p* = 0.3214 in control and +*T.s*.). However, changes in the occurrence of crescent-shaped nuclei were noted during the pachytene stages, unlike the control group, which did not exhibit such changes. This implies that the flawed transition from mitosis to meiosis persisted from the TZ to pachytene stages following treatment with the herb extracts (1.2 vs. 2.2 µm in control and +*T.s*., *p* = 0.0023 by two-tailed Mann–Whitney test).

In line with the larval lethality exhibited in *T.s*. extracts-exposed worms, we observed defective mitotic progression in mitotic gut cells (Figure 2A). In particular, extracts from *Torenia species* frequently triggered the formation of a chromatin bridge, characterized by a strand of chromatin connecting two segregating chromosomes during anaphase or linking daughter nuclei during cytokinesis. Conversely, the untreated control group did not display such a chromatin bridge. This observation suggests the inability to resolve replication or recombination intermediates [11,12].

In the diakinesis stage, six bivalents connected by chiasmata, representing six pairs of homologous chromosomes, are typically observable [3]. Nevertheless, exposure to *T.s*. extracts resulted in the failure to form six pairs of homologous chromosomes, leading to the presence of five DAPI-stained bodies (Figure 2A,C) For six DAPI-stained bodies, 92 vs. 33% in control and +*T.s.*; for five DAPI-stained bodies, 8 vs. 58% in control and +*T.s.*, respectively [13].

Considering the diminished survival observed with the herbal extracts, we proceeded with additional tests to ascertain whether the impaired germline progression led to a decreased brood size. We quantified the progeny number produced by individual hermaphrodite worms beginning from the L4 stage for four days. Our findings revealed a notable decrease in fertility associated with *T.s.* extracts, providing further evidence for defective meiotic development (Figure 2D, 148 vs. 38 in the control and +*T.s.*, *p* < 0.0001). These observations demonstrate how herbal extracts disrupt the systematic advancement of the germline, ultimately resulting in diminished fertility.

This is also supported by the presence of chromatin bridges in mitotic gut cells, indicating defective mitotic progression, as well as the inability to form six pairs of homologous chromosomes during diakinesis suggesting faulty DNA repair (Figure 2A). These cumulative defects ultimately result in a decrease in brood size, highlighting the detrimental effect of herbal extracts on both germline development and fertility. In summary, our observations strongly suggest that *Torenia species* extracts interfere with the normal development of the germline and mitotic cell proliferation, ultimately resulting in compromised fertility.

### 2.5. Torenia species Extracts Activate DNA Damage Checkpoint Pathway

The cellular response to DNA damage is orchestrated by the DNA damage response, which triggers a series of reactions, including DNA damage repair, apoptosis, and cell cycle arrest [14,15,16]. ATM and ATR can activate CHK1, initiating downstream events that facilitate DNA repair, inhibit cell cycle progression, and maintain genome stability in response to DNA damage. We evaluated the mRNA expression of *atm-1* and *atl-1* using qPCR, and the expression of pCHK-1 through immunostaining. Treatment with *Torenia sp*. extracts resulted in an elevated expression of these critical DNA damage checkpoint components, indicating the activation of the DDR following herbal treatment, with a 3.1-fold increase in *atm-1* expression and a 2.0-fold increase in *atl-1* expression (Figure 3A, *p* = 0.00022 and *p* = 0.0011, respectively).

Consistent with the mRNA expression profile, increased levels of pCHK-1 foci were observed in the pachytene stage of germlines (Figure 3B, 1.6 vs. 3.0 in control and *Torenia species*, *p* = 0.0024), confirming the activated DNA damage response following the exposure to herbal extracts [15,17].

Unrepaired DNA intermediates can result in apoptosis in pachytene stage [17]. Hence, we conducted additional investigations into DNA damage-induced apoptosis in the germline. Compared to the untreated control group, which exhibited nuclei rarely highlighted by acridine orange staining, *T.s.* treatments showed an augmentation in the number of nuclei stained with acridine orange during pachytene (Figure 3C, 0.3 vs. 1.2 in control and +*T.s.*, *p* = 0.0008).

Taken together, our results illustrate that exposure to *Torenia species* extracts increases the expression of crucial components in the ATM- and ATR-dependent DNA damage checkpoint pathway, along with elevated levels of phosphorylated CHK-1, indicating the activation of the DNA damage response. These findings strongly suggest the persistence of unrepaired DNA damage, which activates the DNA damage checkpoint, ultimately resulting in heightened apoptosis in the germline.

### 2.6. LC–MS Analysis Identified Anticancer Compounds

In order to gain a deeper understanding of the particular compounds influencing the DNA damage pathway, we performed the LC–MS analysis to pinpoint the active constituents present in *Torenia species* extracts. Worms of the same age were exposed to herb extracts. Subsequently, the samples were homogenized and analyzed using LC–MS. The analysis unveiled the presence of 13 compounds. Among the major components identified were terpenoids and flavonoids (Table 1). The compounds identified included luteolin 7-rutinoside, thymol, dihydrocarvone, cis-carvylacetate, methyl acetate, dihydroisotanshinone I, piceatannol (also known as 4-[2-(3,5-dihydroxyphenyl)ethenyl]benzene-1,2-diol), luteolin, sugiol, aucubin, pilloin, linoleic acid, and homoplantaginin (Figure 4, Table 1, and Appendix A). Ten of them are known for their potential antitumor activity.

## 3. Discussion

We employed *C. elegans* to examine the potential nematocidal toxicity of herbal extracts and to shed light on their roles in DNA damage repair and checkpoint responses (Figure 5). We discovered that *Torenia sp*. is an inducer of DNA damage checkpoint responses, DNA damage apoptosis, aberrant sex chromosome segregation, defective meiotic progression, and reduced survival rates.

### 3.1. Medicinal Potential of Torenia species

*Torenia species*, commonly known as “wishbone flowers” or “blue wings”, are dicotyledonous plants prevalent in tropical and subtropical regions across Asia, Africa, and Madagascar [19]. Some *Torenia species*, including *Torenia fournieri*, *Torenia concolor*, *Torenia asiatica*, and *Torenia hybrida*, contribute to the vibrant array of flower colors seen in gardens and landscapes worldwide. These plants have garnered attention not only for their aesthetic appeal but also for their utility in genetic research, owing to an efficient transformation system [20] and a small genome conducive to genomic studies [21].

Moreover, certain *Torenia species*, such as *T. concolor* Lindley var. formosana Yamazaki (TC), have been traditionally used for treating various human illnesses, including hypertension, stomatitis, hepatitis, pneumonia, and gastroenteritis. These medicinal properties are attributed to the plant’s detoxification, anti-inflammatory, and diuretic effects. Experimental studies have demonstrated the anti-inflammatory effects of TC extracts on macrophages and their ability to inhibit lipid deregulation in adipocytes [22,23]. Additionally, compounds found in *Torenia fournieri* exhibit antioxidative activity in rat brains [24], with luteolin-7-O-beta-glucoside identified as a notable component, akin to Luteolin 7-O-rutinoside. Furthermore, TC extracts have been found to promote nitric oxide production via the Akt/CaMKII/AMPK/eNOS signaling cascade, effectively inhibiting inflammatory responses and impeding the progression of atherosclerosis [25]. Despite these well-documented medicinal properties, the potential antitumor or insecticidal activity of *Torenia species* remains relatively unexplored, representing an area ripe for systematic investigation.

### 3.2. Potential Anticancer Properties of Torenia species

Anticancer compounds found in the *T.s.* extracts contribute to the phenotypes exhibited in *C. elegans*. For instance, linoleic acid, a vital omega-6 polyunsaturated fatty acid found in both *Torenia species* and *Onobrychis cornuta*, recapitulated the phenotypes exhibited in the herb extracts, which include increased levels of pCHK-1 foci, apoptosis, and the activation of the MAPK pathway [7].

It is important to highlight that the majority of compounds found in *Torenia species* extracts are also matched to those found in extracts of *Onobrychis cornuta* (*O.c.*) and *Veratrum lobelianum* (*V.l.*), as identified in our previous study activation [7]. Specifically, 12 out of 13 findings in *Torenia species* extracts match those in *O.c.*, and 9 out of 13 findings in *Torenia species* extracts were found in both *O.c.* and *V.l.* This intriguing correlation underscores the potential for *Torenia species* to possess similar antitumor properties as *O.c*. and *V.l.* extracts. Furthermore, in line with this idea, all three herb extracts exhibited a similar phenotype of the induced apoptotic pathway triggered by DNA damage.

The presence of the majority of compounds in extracts from *Torenia species*, which are also found in extracts from *O.c.* and *V.l.*, suggests a potential commonality in their biochemical composition and biological activities. This correlation raises questions about the underlying mechanisms through which these compounds exert their effects, prompting further exploration of the specific compounds shared between these plant species and their roles in inducing apoptotic pathways triggered by DNA damage. Furthermore, it highlights the importance of studying the interactions between these compounds and their targets within the cellular environment to elucidate their therapeutic potential and optimize their use in clinical applications.

Excitingly, *Torenia species* also contain a unique polyphenolic compound known for its pro-apoptotic and antitumor activity, a feature not found in the other two herbs. Piceatannol, a structural analog of resveratrol found in grapes and berries, is known to induce caspase-dependent apoptosis, nucleosomal DNA fragmentation, and PARP1 cleavage in many human cancer cell lines, including leukemia, melanoma, and lymphoma [26,27].

It is surprising that *Torenia species*, frequently known for its ornamental beauty, contains numerous antitumor compounds. Further investigation into the shared compound composition and observed phenotypes among these extracts could provide valuable insights into their mechanisms of action and potential applications in cancer treatment. Additionally, exploring the specific pathways and molecular targets affected by these compounds may elucidate the underlying mechanisms driving their antitumor effects, ultimately contributing to the development of novel therapeutic strategies.

### 3.3. Conclusions

In this study, we investigated the nematocidal attributes of *Torenia species*, known for their esthetic appeal. Our findings revealed that these extracts caused a decrease in survival and disrupted mitotic and meiotic developmental processes in *C. elegans*. These phenotypes were linked to the activation of DNA damage checkpoint and apoptosis. Notably, our analysis uncovered that the extracts contained compounds with established antitumor activity, suggesting potential pharmaceutical applications beyond traditional gardening. Particularly noteworthy was the discovery of compounds like piceatannol, which possess known pro-apoptotic and antitumor properties, highlighting the hidden therapeutic potential of these ornamental plants. Further investigation into the shared compound composition and observed phenotypes among extracts could provide valuable insights into their mechanisms of action, potentially paving the way for the development of novel cancer treatment strategies.

## 4. Materials and Methods

### 4.1. Strains and Alleles

All *C. elegans* strains were cultured under standard conditions at 20 °C, and the N2 Bristol strain was employed as the wild type, as described by Brenner, S. [28].

### 4.2. Herb Extraction

*Torenia sp*. was gathered in Armenia in May 2006. The plant material was purified to remove extraneous matter, air-dried in the shade, ground into a coarse powder, and subsequently extracted with methanol. The combined methanol extract was evaporated under vacuum to afford a tarry residue. The methanol extract was mixed with 90% (aqueous) methanol and subjected to extraction with n-hexane.

The residual hydroalcoholic phase was stripped of the solvent under vacuum, suspended in water, and then subjected to sequential extraction with dichloromethane and n-butanol. This process yielded a rough separation into hexane-, butanol-, dichloromethane-, and water-soluble fractions.

The herbal extracts in the solvents hexane, butanol, and water, labeled as -H, -B, and -A, respectively, were dissolved in DMSO. Following their initial preparation, the extracts were diluted to a concentration of 1 mg/mL in DMSO and stored in a freezer. For subsequent assays, they were further diluted using M9 buffer to achieve a final concentration of 0.03 µg/mL.

### 4.3. Larval Arrest or Lethality, Survival, Brood Size, and HIM

Gravid hermaphrodites were obtained from NGM plates to synchronize L1 stage larvae, following the procedure described by Kim, H.M. et al. [15,29]. The synchronized worms were subsequently immersed in 180 µL of the herb extract solution and transferred to a 96-well plate. Afterward, the worms were gently agitated at 10–30 RPM and incubated at 20 °C for 24 h. Phenotypic changes were continuously monitored for up to 48 h.

In assessing their relative survival, the mobility of the worms was monitored following 24 h of incubation. Brood sizes were determined by tallying the cumulative number of eggs laid by individual worms during the 4–5 days post-L4 stage. Larval arrest/lethality was calculated as the % of hatched worms that failed to reach adulthood. The proportion of males in the population was quantified to ascertain the percentage of adult worms that were males. All experiments were conducted in triplicate to ensure reproducibility. The survivability study protocol was adapted from the work of Kim and Colaiacovo [29].

### 4.4. Worm Lysates for Mass Spectrometry

Worm preparation methods for LC–MS analysis were outlined by Meng et al. [7]. Worms of the same age (20 h post-L4) were exposed to M9 buffer containing the herb extracts (0.03 µg/mL) at 25 °C for 20 h. Following exposure, the worms underwent 10 washes in M9 buffer and were then rapidly frozen with minimal M9 in liquid nitrogen. The frozen worm pellet was subsequently suspended in lysis buffer (0.5 M sucrose, 25 mM HEPES (pH 7.6), 5 mM EDTA, 0.5% CHAPS, 0.5% deoxycholic acid). The samples underwent homogenization and centrifugation to remove worm debris. MS analysis was conducted by YanBo Times (Beijing, China).

### 4.5. LC–MS Analysis

This study employed a LC-30A chromatography (Shimadzu, Kyoto, Japan) with a C18 column (2.1 × 100 mm, 2.2 μm) by Yanbotimes (Beijing, China). The column temperature was kept at 40 °C, while the flow rate was adjusted to 0.2 mL/min. A 2 μL sample was injected onto the column. The mobile phase comprised acetonitrile and a 0.1% formic acid solution. We employed the AB Sciex Triple TOF 5600+ for mass spectrometry analysis.

In positive ionization mode, the ion source voltage was set to 5500 V, and the ion source temperature was maintained at 500 °C. The declustering potential was adjusted to 100 V, collision energy to 40 eV, and collision energy spread to 15 eV. Nitrogen gas was utilized as the nebulizing gas, with pressures set to 50 PSI for both auxiliary gases 1 and 2, and 40 PSI for the curtain gas. The mass spectrometer conducted first-level MS scanning across the range of 100–1500 m/z, followed by second-level MS scanning for peaks with a response exceeding 100 cps. The second-level MS scanning was set to cover the range of 100–1500 m/z, with dynamic background subtraction enabled.

In negative ionization mode, the ion source voltage was adjusted to −4500 V, remaining consistent with all other parameters set in the positive ionization mode. The obtained results were cross-referenced with the company’s standard sample database.

### 4.6. Monitoring the Growth of E. coli

The growth behavior of *E. coli* OP50 was assessed using optical density measurements, following the methodology outlined by Meng et al. [7,30]. In brief, bacterial growth was monitored by regularly measuring optical density at 600 nm while exposing the bacteria to each herb extract at a concentration of 0.03 µg/mL.

### 4.7. Immunofluorescence Staining

Whole-mount preparations of dissected gonads, along with the fixation and immunostaining procedures, were performed according to the methods described by Kim and Colaiacovo [15,17,31]. Primary antibodies were utilized at the dilutions provided below: rabbit pCHK-1 (1:250, Cell Signaling, Danvers, MA, USA, Ser345). The secondary antibodies used were Cy3 anti-rabbit (1:300) from Jackson Immunochemicals. Immunofluorescence images were obtained at intervals of 0.2 μm using an Eclipse Ti2-E inverted microscope and a DSQi2 camera (Nikon, Konan, Japan). Photos were taken with a 60× objective lens, enhanced by 1.5× auxiliary magnification, and underwent deconvolution using NIS Elements software Version 3 (Nikon). The images depict partial projections of half nuclei.

### 4.8. Quantitative Analysis of pCHK-1 Foci

Quantification of pCHK-1 foci was conducted following the method outlined by Kim and Colaiacovo [15,17]. Between five and ten germlines were evaluated for each treatment. Statistical comparisons between treatments were carried out using either the two-tailed Mann–Whitney or T-test with a 95% confidence interval.

### 4.9. Quantitation of Germline Apoptosis

Germlines of animals matched in age (20 h post-L4) were examined using acridine orange staining, following the protocol outlined by Kelly et al. [32], and observed under a Nikon Ti2-E fluorescence microscope. Between 20 and 30 gonads were assessed for each treatment. Statistical analyses comparing treatments were conducted using the two-tailed Mann–Whitney test, with a 95% confidence interval.

### 4.10. Quantitative Real-Time PCR

cDNA was synthesized from the RNA extracts of young hermaphrodite worms using the AB-script II First Strand cDNA Synthesis Kit (ABclonal RK20400). Real-time quantitative PCR (qPCR) amplification was performed using ABclonal 2X SYBR Green Fast mix (Abclonal RK21200). The amplification process was carried out in a LineGene 4800 instrument (FQD48A BIOER) with an initial polymerase activation step at 95 °C for 2 min, followed by approximately 40 cycles consisting of denaturation at 95 °C for 15 s, annealing at 60 °C for 20 s, and elongation at 60 °C for 20 s. Following amplification, a melting curve analysis (60 °C–95 °C) was conducted to confirm the specificity of the amplicons. Tubulin encoding *tba-1* was chosen as the reference gene based on *C. elegans* microarray expression data. Each qPCR assay was independently repeated at least twice.

## Figures and Tables

**Figure 1 pharmaceuticals-17-00611-f001:**
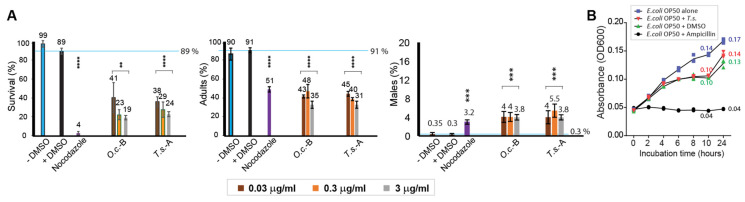
**Extracts derived from *Torenia species* demonstrate a significant inhibitory effect on both the survival and developmental progression of worms, without exerting any discernible impact on bacterial growth.** (**A**) The impact of *Torenia sp*. extracts on survival and development reveals significant nematocidal effects. The survival percentage, adult percentage, and HIM percentage of *C. elegans* were monitored after a 24 h treatment with the herb extract and observed for 48 h. Survival and adult formation showed an inverse correlation with increasing doses of the herbal extract, while the percentage of males was not affected. *Torenia sp*. extracts were treated at final concentrations of 0.03, 0.3, and 3 µg/mL. Blue horizontal lines indicate the value of negative control (+DMSO). Statistical significance between +DMSO and the samples is indicated by asterisks. ** *p* ≤ 0.01; *** *p* ≤ 0.001; **** *p* ≤ 0.0001. (**B**) An evaluation of the herb extracts’ impact on bacterial growth showed no significant bacterial growth defect over 24 h indicating that the nematocidal effects were not primarily due to compromised bacterial growth. *T.s*-H was added to a final concentration of 0.03 μg/mL.

**Figure 2 pharmaceuticals-17-00611-f002:**
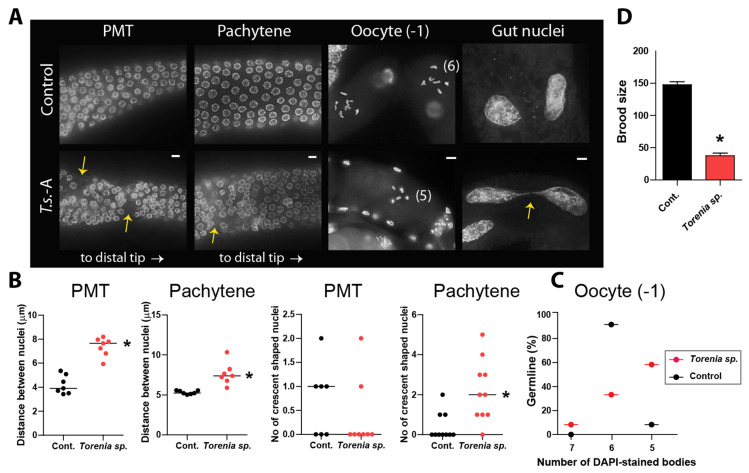
**Aberrant germline development induced by *Torenia sp*.-H extracts.** (**A**) Nuclei arrangement during germline development. Exposure to *Torenia species* (*T.s.*-H) extracts led to the emergence of enlarged gaps, as denoted by arrows, between nuclei observed in both the premeiotic tip and pachytene stages. These gaps were notably more pronounced in worms treated with the herb extracts. Additionally, worms exposed to the herb extracts exhibited a decrease in the number of DAPI-stained bodies during diakinesis, suggesting a compromised DNA recombination process. Furthermore, mitotic gut nuclei in these worms displayed chromatin bridges, highlighted by arrows, further indicating aberrations in cellular division processes. Scale bar = 2 µm. (**B**) Left panel: Quantification of the increased distance between the nuclei of premeiotic tip (PMT) and pachytene stages shown in (**A**). Right panel: Number of crescent-shaped nuclei induced in pachytene of *T.s.*-exposed worms. (**C**) Quantification of DAPI-stained bodies in the germline. The percentage of germlines containing DAPI-stained bodies is indicated. (**D**) Brood size of *T.s.*-H-exposed worms. Exposure to *Torenia sp*. extracts led to a notable decrease in the progeny count produced by hermaphroditic worms over a four-day period. Asterisks denote statistical significance as determined by the two-tailed Mann–Whitney test. Data are expressed as the mean ± SEM.

**Figure 3 pharmaceuticals-17-00611-f003:**
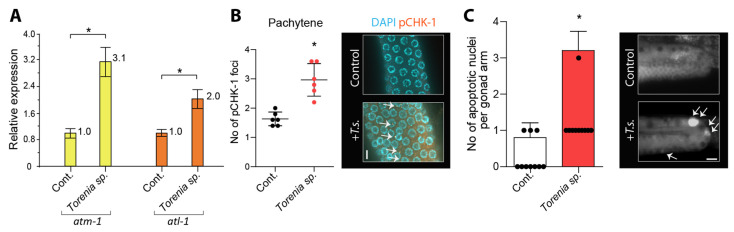
**Exposure to herbal extracts of *Torenia sp*. induces the active DNA damage response pathway in *C. elegans*, leading to elevated levels of key DNA damage checkpoint components, phosphorylated CHK-1, and apoptosis.** (**A**) Exposure to *Torenia sp.*-H extracts results in heightened expressions of *atm-1* and *atl-1*, confirming the activation of the DNA damage response pathway. (**B**) Elevated pCHK-1 foci observed in germlines underscores the active DNA damage response following exposure to *T.s.*-H extracts. Arrows indicate pCHK-1 foci localized on nuclei. Bar = 2 µm. (**C**) The pachytene stage demonstrated an elevated incidence of apoptosis, indicating that unrepaired DNA damage prompts checkpoint activation and subsequent apoptosis (arrows) within the germline. Asterisks indicate statistical significance by the two-tailed Mann–Whitney test. Bar = 20 µm.

**Figure 4 pharmaceuticals-17-00611-f004:**
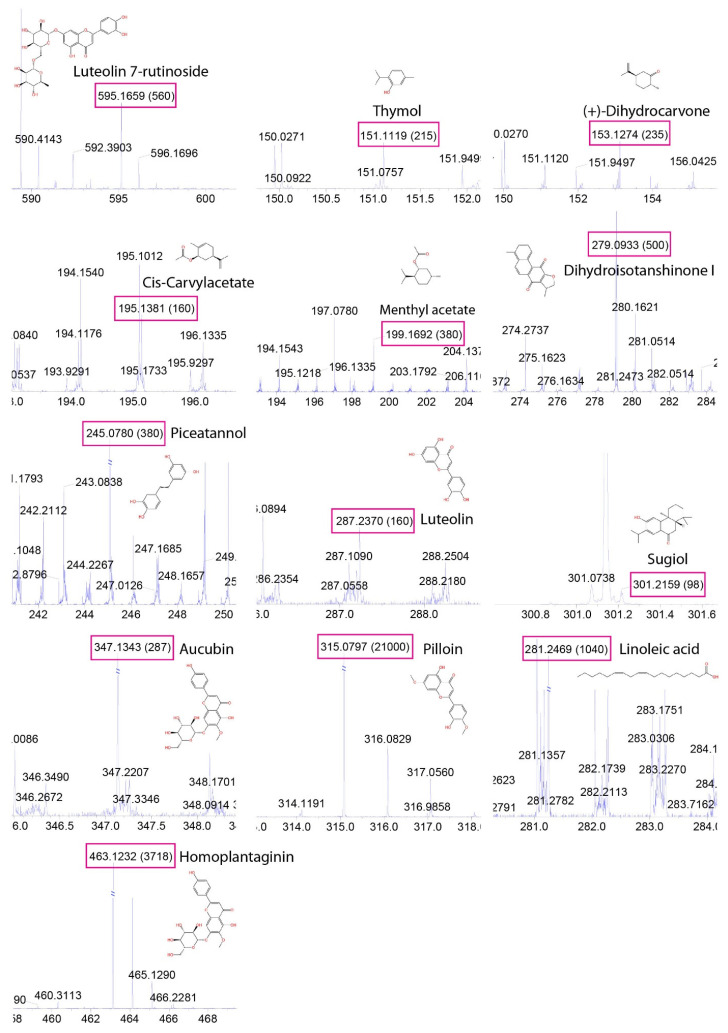
**Spectrum of phytochemicals identified in *Torenia sp*.** Thirteen compounds were found in herb extracts (*T.s.*-H). The predicted fragmentation of compounds, as indicated by red boxes, was provided by Analyst 1.6, an in silico analysis tool. The *x* axis in an LC–MS graph represents the mass-to-charge ratio (m/z), indicating the size and charge of ions. The numbers enclosed in brackets denote the intensity in counts per second (CPS), indicating the quantity of ions of a specific m/z detected.

**Figure 5 pharmaceuticals-17-00611-f005:**
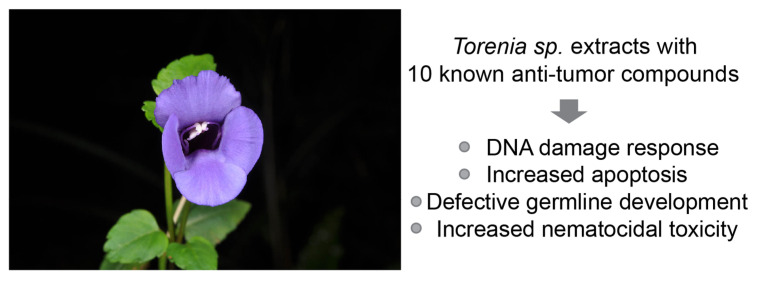
***Torenia sp.* extract induces germline abnormalities and influences worm survival.** *Torenia sp*. extracts trigger a DNA damage response pathway, resulting in impaired germline development and increased apoptosis, and interfere with meiotic progression. This underscores the potential cytotoxicity of *Torenia sp*. extracts. Furthermore, the presence of compounds with known antitumor activity in *Torenia sp*. extracts, including those shared with *Onobrychis cornuta* and *Veratrum lobelianum*, suggests the possibility of utilizing *Torenia species* in anti-cancer therapeutics beyond traditional gardening applications. Photographed by Nickrent et al. (http://www.phytoimages.siu.edu, accessed on 20 April 2024, [18]).

**Table 1 pharmaceuticals-17-00611-t001:** **Phytochemical composition of *Torenia sp*. The types of phytochemicals were highlighted with different colors, with the CAS number of each compound designated in parentheses.** Due to limitations in the LC–MS strategy we employed, we are unable to detect the quantity of individual compounds. Nevertheless, all components have a concentration exceeding 0.0003 g/kg or 0.003 g/L, with this being the lowest concentration observed.

No	Compound (CAS)	Rt	[M + H]+	Fragments	Types
1	**Luteolin 7-O-rutinoside (** **20633-84-5** **)**	11.648	595.1659	286.0484, 301.0728	Flavonoids
2	**Thymol (** **89-83-8** **)**	20.39	151.1119	77.0414, 91.0553, 105.0705	Terpenoids
3	**(+)-Dihydrocarvone (** **7764-50-3** **)**	22.663	153.1274	152.1122	Terpenoids
4	**Cis-Carvyl acetate (** **97-42-7** **)**	23.221	195.1381	77.0426, 91.0583	Terpenoids
5	**Menthyl acetate (** **89-48-5** **)**	27.985	199.1692	197.0476, 69.0747	Terpenoids
6	**Dihydroisotanshinone I (** **87205-99-0** **)**	19.354	279.0933	149.0239	Tanshinones
7	**Piceatannol (** **10083-24-6** **)**	21.001	245.078	108.9963, 140.9179	Analog of resveratrol
8	**Luteolin (** **491-70-3** **)**	27.589	287.237	145.1026	Flavonoids
9	**Sugiol (** **511-05-7** **)**	23.15	301.2159	259.1685, 163.0752	Terpenoids
10	**Aucubin (** **479-98-1** **)**	26.472	347.1343	216.9976, 129.0189	Iridoids
11	**Pilloin (** **32174-62-2** **)**	1.121	315.0797	182.0427	Carotenoid
12	**Linoleic acid (** **60-33-3** **)**	31.101	281.2469	265.0125, 150.0269	Fatty acids
13	**Homoplantaginin (** **17680-84-1** **)**	12.293	463.1232	301.0724, 286.0489	Alkaloids

## Data Availability

Data are contained within the article and Appendix A.

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
