# Peer review of "Torenia sp. Extracts Contain Multiple Potent Antitumor Compounds with Nematocidal Activity, Triggering an Activated DNA Damage Checkpoint and Defective Meiotic Progression"

_pharmaceuticals, 2024, doi:10.3390/ph17050611_

Round 1

Reviewer 1 Report

Comments and Suggestions for Authors

Comments:

It is an interesting work with data that is relevant but some corrections need to be made;

The title must correspond to the results.

1.- The summary must include the compounds or type of compounds that were found.

2.- In the introduction they repeat paragraphs that are in the summary. They should only focus on the results of this work in the summary, I propose that it be removed from the summary and left in the introduction. Biological and chemical activities of the Torenia species should also be included in the introduction.

3.- Add to table 1 the amount of each one in the extract.

4.- It is necessary to present the identification number of the species under study

5.- In the methodology, the methanolic extract is separated to obtain the fractions with hexane, dichloromethane, butanol and water. What is the objective if they were not evaluated? What fractions did LC-MS do? Present chemistry data of the fractions. 

Reviewer 2 Report

Comments and Suggestions for Authors

The manuscript provides a concise overview of a study investigating the nematocidal properties of herbal extracts, with a specific focus on Torenia sp. The study examined 316 herbal extracts, revealing that 16% of them demonstrated nematocidal effects on Caenorhabditis elegans, including diminished survival rates and larval arrest. The authors report the effects of Torenia sp., highlighting its ability to decrease survival rates and increase larval arrest/lethality. Additionally, it discusses the impact of Torenia sp. on developmental processes, including perturbed meiotic advancement and activation of DNA damage checkpoint responses. The study also identifies 13 compounds within Torenia sp. extracts, 10 of which are recognized for their antitumor activity, suggesting potential therapeutic applications beyond gardening. Overall, the text provides valuable insights into the potential therapeutic properties of herbal extracts, particularly Torenia sp., and hints at future research directions in the field.

The article holds significant relevance in light of humanity's ongoing battle against all forms of cancer.

I like the introduction part. It focusses mainly of the previous work of the authors and how they continue their work, introducing the results in this manuscript.

The discussion part is can be extended with detailed discussion of the results. Also, please include proper conclusion of this part.

The herb was collected in Armenia in May 2006. This is 18 years ago. I wonder when was carried the extraction work? Back in 2006 or recently before the publication? Please explain how was kept the material during these years if the analysis was performed recently.

Row 330 Worm preparation methods for LC‒MS analysis were described in [7]. Please revise. For example ……….were described by Meng et al. [7]. Please revise all manuscript about the same issue.

All methods are described correctly, and the results are presented in professional manner.

Please make sure you update the references regarding the journal style.

Regarding my comments above, I would like to recommend minor revision before publication.  

Round 2

Reviewer 1 Report

Comments and Suggestions for Authors

 None

Reviewer 2 Report

Comments and Suggestions for Authors

I thank the authors for considering my recommendations and criticisms. I think their article is much better presented now.